# Lipoprotein Particles as Shuttles for Hydrophilic Cargo

**DOI:** 10.3390/membranes13050471

**Published:** 2023-04-28

**Authors:** Florian Weber, Markus Axmann, Andreas Horner, Bettina Schwarzinger, Julian Weghuber, Birgit Plochberger

**Affiliations:** 1Department of Medical Engineering, University of Applied Sciences Upper Austria, 4020 Linz, Austria; florian.weber2@fh-linz.at (F.W.); markus.axmann@fh-linz.at (M.A.); 2Science for Life Laboratory, Department of Women’s and Children’s Health, Karolinska Institutet, 17164 Solna, Sweden; 3Institute of Biophysics, Johannes Kepler Universität, 4040 Linz, Austria; andreas.horner@jku.at; 4FFoQSI—Austrian Competence Centre for Feed and Food Quality, Safety & Innovation, 4600 Wels, Austria; bettina.schwarzinger@fh-wels.at (B.S.); julian.weghuber@fh-wels.at (J.W.); 5Center of Excellence Food Technology and Nutrition, University of Applied Sciences Upper Austria, 4600 Wels, Austria

**Keywords:** lipoprotein particles, shuttle for hydrophilic cargo, insulin, single-molecule-sensitive fluorescence microscopy, atomic force microscopy

## Abstract

Lipoprotein particles (LPs) are excellent transporters and have been intensively studied in cardiovascular diseases, especially regarding parameters such as their class distribution and accumulation, site-specific delivery, cellular internalization, and escape from endo/lysosomal compartments. The aim of the present work is the hydrophilic cargo loading of LPs. As an exemplary proof-of-principle showcase, the glucose metabolism-regulating hormone, insulin, was successfully incorporated into high-density lipoprotein (HDL) particles. The incorporation was studied and verified to be successful using Atomic Force Microscopy (AFM) and Fluorescence Microscopy (FM). Single-molecule-sensitive FM together with confocal imaging visualized the membrane interaction of single, insulin-loaded HDL particles and the subsequent cellular translocation of glucose transporter type 4 (Glut4).

## 1. Introduction

Non-water-soluble substances, such as fats from storage sites or food, are transported in the blood stream in the form of lipoprotein particles (LPs). LPs are micelle-like (i.e., only one lipid leaflet) transporters in higher organisms consisting of proteins (i.e., apolipoproteins) associated with lipids and other substances [1,2]. Their envelope surrounds a central lipophilic core of triglycerides and cholesterol esters. LPs can be divided according to their density into high-density, low-density, and very low-density LPs (HDL, LDL, and VLDL). Briefly, LDL particles transport cholesterol to cells [3,4,5], and excess cholesterol is absorbed by HDL particles and transported to the liver [6,7,8]. An imbalance between LPs may lead to an increased systemic cholesterol level and triggers inflammatory processes in blood vessels [9,10]. Thus, HDL particle functionality is a promising approach for treatment of cardiovascular diseases [11,12]. Specifically, HDL particles interact with certain cell types involved in lipid metabolism by binding to receptors, such as the scavenger receptor class B type 1 (SRB1) [13,14,15,16], and also to the LDL receptor (LDL-R) [17] or adenosine triphosphate-binding cassette transporters of subfamily G, member 1 (ABCG1) [16]. Some of these receptors are also overexpressed on cancer cells and atherosclerotic plaques. Consequently, LPs, especially HDLs and LDLs, represent a promising tool for drug delivery [18,19]. Moreover, LPs can also be used as diagnostic markers by analyzing, e.g., their miRNA cargo [20]. Due to their high turnover rate, their physiologically controlled metabolism, and their stability in the blood plasma, LPs are perfectly suited as shuttles for the transfer of active lipophilic substances [21,22,23,24] or essential micronutrients such as vitamins [25]. Natural and synthetic LPs have been surveyed in nanomedicine for drug release owing to the high biocompatibility, tunable carriers, as well as easy surface modification and functionalization [26]. As reviewed by Tsujita et al. [27], synthetic nanodisc (i.e., disc-shaped bilayers stabilized by a scaffold (apolipo)protein) HDL-like particles have been used in a variety of applications as drug carriers for amphipathic/hydrophobic molecules. Similarly, reconstituted HDL (rHDL) particles generated from an aqueous mixture of apolipoprotein(s) and lipids can be used as therapeutic delivery devices (for a review, see Fox et al. [28]) via drug incorporation into these particles’ core or their lipid monolayer via His-tagged proteins and nickel-chelating lipids [29]. In addition to LPs, artificial lipoplexes can be used to transfer, e.g., small interfering RNA (siRNA) for gene silencing via His-tagged targeting proteins, which are incorporated into the membrane of the lipoplexes via NTA-tagged lipids [30]. LPs are also able to shuttle small drug molecules across the blood–brain barrier for treatment of brain diseases. Hence, insulin receptor and low-density lipoprotein (LDLR) are known key receptors expressed in the blood–brain barrier that mediate the shuttling of macromolecules [31,32].

This work examines the successful incorporation of an active substance within HDL particles and investigates their delivery as well as physiological effect. In particular, we were able to load HDL particles with insulin, which was linked to their envelope by means of a lipid-anchored Ni-NTA/His-tagged insulin complex. Using Fluorescence and Atomic Force Microscopy (FM/AFM), the insulin-loaded HDL particles were investigated with respect to their shape, interaction with biomembranes, and stimulation effect on living cells.

## 2. Materials and Methods

### 2.1. Reagents and Cells

Reagents: Sodium cyanoborohydride (NaCNBH_3_), CHCl_3_, CH_3_OH, CaCl_2_, NaCl, TriEthylAmine (TEA), 3-AminoPropyl-TriEthoxySilan (APTES), EthanolAmine (ETA), sodium deoxycholate, sucrose, glucose, 4-(2-HydroxyEthyl)-1-PiperazineEthaneSulfonic acid (HEPES), phosphate buffered saline (PBS), Hanks’ balanced salt solution (HBSS), Ham’s F12 culture medium, paraformaldehyde (PFA), dithiothreitol (DTT), penicillin, streptomycin, G418, and fetal bovine serum (FBS) were obtained from Sigma-Aldrich(Vienna, Austria). 1,2-DiOleoyl-sn-glycero-3-PhosphoCholine (DOPC) and 1,2-dioleoyl-sn-glycero-3-[(N-(5-amino-1-carboxypentyl) iminodiacetic acidsuccinyl] (nickel salt) (18:1 DGS NTA(Ni)) were obtained from Avanti Polar Lipids (Birmingham, AL, USA). Insulin polyclonal antibody Alexa Fluor 488 conj was obtained from Bioss Antibodies Inc. (THP Medical Products, Vienna, Austria). Human insulin (recombinant, His_6_, and N-terminus) was obtained from LSBio (eubio, Vienna, Austria). Biotin-insulin was purchased from Ibt (immunological and biochemical testsystems, Binzwangen, Germany). Alexa Fluor 647 NHS and streptavidin was purchased from Thermo Fisher Scientific (Vienna, Austria).

Cells: CHO-K1 cells stably expressing human insulin receptor and GLUT4-myc-GFP (i.e., CHO-K1-hIR/GLUT4-myc-GFP) have been described in previous studies [33,34]. Briefly, these cells do not need to be differentiated, are highly sensitive to insulin, and have been well characterized regarding the concentration of insulin receptor present in the plasma membrane. The cells were grown in a humidified atmosphere (≥95%) at 37 °C and 5% CO_2_ and maintained in Ham’s F12 culture medium supplemented with 100 µg/mL of penicillin, 100 µg/mL of streptomycin, 1% G418, and 10% FBS. The cells were grown on clean glass slides (Ø 15 mm) for 2 days. Before measurements, the cells were starved using Hanks’ buffered saline solution (HBSS) without Ca/Mg for 2 h. As a result, the background/autofluorescence could be reduced by about 20%.

### 2.2. Lipoprotein Particle Isolation and Labeling

Blood donations, which were obtained from normolipidemic healthy volunteers, were approved by the Ethics Committee, Medical University of Vienna (EK-Nr. 511/2007, EK-Nr. 1414/2016). Lipoprotein particles were isolated as previously described [35] via sequential flotation ultracentrifugation. Briefly, HDL particles were isolated from less dense lipoprotein particles and denser proteins due to the fact that they float at a density range of 1.063–1.210 g/mL. The overall solution density was sequentially increased using KBr, and these particles were separated using ultracentrifugation. The particles’ proteins were covalently linked to Alexa Fluor 647 at a pH = 8.3 according to the manufacturer’s instruction. Free dye was removed via extensive dialysis [36].

### 2.3. Insulin Loading of HDL Particles

A total of 50 µL of 18:1 DGS NTA (Ni) (C_DGS_ = 1 mg/mL) dissolved in CHCl_3_ were dried in a clean glass tube under N_2_ gas flow while rotating the tube to gain a homogeneous lipid layer. Then, 500 µL of HDL particle solution (C_protein_ = 2 mg/mL) was added to the lipid-containing glass tube and incubated for 30 min at 37 °C. Free lipids were removed via a two-step sequential flotation ultracentrifugation according to the protocol described above (Section 2.2). The purified HDL-18:1 DGS NTA (Ni) particles were incubated with 18 µL of 110 µg/mL His_6_-insulin for 1 h at room temperature. Free His_6_-insulin was removed via PD-10 desalting columns.

### 2.4. Immobilization of HDL Particles for Imaging via Confocal Microscopy and Force Spectroscopy Imaging

Two approaches were chosen to control the density of the immobilized HDL particles. First, the glass surface was coated with APTES. In the first case, the HDL particles loaded with 18:1 DGS NTA (Ni) lipid were incubated directly onto the APTES surface, whereas in the second case, the particles were positioned via micro-contact printing [37], with a concentration of 100 µg/mL in both cases. By using the second approach, the interaction surface with HDL particles was reduced by half for subsequent exchange studies and, hence, led to a decrease in transferable lipids as well. After HDL particle immobilization, insulin-His_6_ was added to the immobilized HDL particles, and then unbound insulin was removed by multiple washing steps. To reveal the bound insulin molecules microscopically, a fluorescently labeled antibody against insulin was used (see Appendix A which shows the two functionalized surfaces of homogeneous versus structured HDL particles). For the transfer experiments, an anti-insulin antibody was added after GUV contact in order to visualize transferred insulin within the target membrane.

For the force spectroscopy experiments (Appendix A), force cycles were acquired using silicon cantilevers with a spring constant of 0.2 N/m (NanoAndMore, Wetzlar, Germany —CP-CONT-BSG) and a spherical tip with a nominal radius between 5 and 20 µm. The AFM probes were incubated with 30 µL (1 mg/mL) of streptavidin and subsequently washed by rinsing with PBS; afterward, they were incubated with biotin-insulin at a concentration of 100 µg/mL for 30 min. After incubation, the cantilevers were washed by rinsing with PBS. The insulin-coated spherical tip was moved toward the cell surface of CHO-K1-hIR/GLUT4-myc-GFP. The tip was held in contact at a constant force, and after 20 min, it was again retracted. Simultaneously, HILO microscopy [38] was used to visualize the response of the labeled GLUT4 transporters on the cell membrane.

### 2.5. Atomic Force Microscopy (AFM) Imaging and Particle Analysis

In agreement with previously published works of other groups [39,40,41,42,43], the AFM measurements of HDL particles were adapted for our Atomic Force Microscope (JPK BioAFM—NanoWizard 4, JPK, Berlin, Germany). AFM probes made of silicon nitride with a nominal spring constant of 0.25 N/m and a nominal tip radius between 5 and 8 nm (FASTSCAN-D, Bruker Nano Inc., Camarillo, CA, USA) were used. The exact sensitivity and the spring constant of each cantilever were determined on a clean coverslip in 300 µL of PBS based on a force–displacement experiment and a thermal noise spectrum measurement [44]. Native and insulin-loaded HDL particles were diluted 1:1000. A total of 300 µL of the diluted HDL particle solution were incubated on the clean glass coverslip for at least 5 min and subsequently imaged. The AFM images were obtained by using an advanced imaging mode (Quantitative Imaging mode: QI™ mode) of Bruker. The maximal force was set to 500 pN. Particle analysis (Full Width at Half Maximum (FWHM) and height) was performed with the JPK Data Processing software (V.6.1.163, JPK, Berlin). For an estimation of the central tendency of the sample heights, a model of the data density distribution was performed using “fitdist” (MatLab, Version R2020a, Mathworks, Natick, MA, USA). The model was interpreted with a probability density function using “pdf” ”(MatLab, Version R2020a, Mathworks, Natick, MA, USA) [45]. To test whether the height distribution of the different samples originated from the same population, the nonparametric Kruskal–Wallis test was performed [46]. This test is an extension of the Wilcoxon rank-sum test. 

### 2.6. Cell Stimulation and Giant Plasma Membrane Vesicle (GPMV) Formation

CHO-K1-hIR/GLUT4-myc-GFP cells were grown in a 96-well plate and incubated with the test substances (HBSS, 100 nM of soluble insulin, 125 µg/mL of HDL, and 125 µg/mL of insulin-loaded HDL) for 10 min. GPMV buffer (10 mM of HEPES, 2 mM of CaCl_2_, and 150 mM of NaCl, at a pH of 7.4) and GPMV-forming buffer (25 mM of PFA, 2 mM of DTT, 10 mM of HEPES, 2 mM of CaCl_2_, and 150 mM of NaCl, at a pH of 7.4) were freshly prepared. The cells were washed three times with the GPMV buffer [47] and incubated in the GPMV-forming buffer for 1 h. All incubation steps were performed in a humidified atmosphere (≥95%) at 37 °C and 5% CO_2_. Finally, the buffer with the formed GPMVs was transferred into the empty wells of a 96-well imaging plate [48].

### 2.7. Confocal Microscopy of GPMVs and Giant Unilamellar Vesicle (GUV)

GPMVs were imaged with a laser scanning confocal microscope (LSM 700 AxioObserver, Zeiss, Vienna, Austria). The microscope was equipped with a Plan-Apochromat 63×/1.40 Oil DIC M27 objective (Zeiss, Vienna, Austria). The LSM 700 operates with solid lasers (polarization-preserving single-mode fibers) at wavelengths of 639 nm and 488 nm. Signals were detected after appropriate filtering on a photomultiplier. Detector amplification, laser power, and pinhole were kept constant for all measurements. The images were analyzed by thresholding the signal; the average pixel value of the signals underneath the obtained mask, which followed the GPMV surface, was calculated in the respective GFP channel.

GUVs were imaged with an inverted laser scanning confocal microscope (LSM510META Confocor3, Zeiss, Vienna, Austria) equipped with a water-immersion objective (Plan-Apochromat, 40×, NA 1.2, Zeiss, Vienna, Austria). Equatorial images of GUVs were recorded at a wavelength of 488 nm via a dichroic beam splitter (488/561/633). The fluorescent signal was detected by means of an avalanche photodiode using a band-pass filter (BP 505–610 nm). The images were analyzed using the Zeiss ZEN V3.6 software with ImageJ V1.53q. DOPC-GUVs were formed using DOPC solution (10 mg/mL in CHCl_3_:CH_3_OH, 3:1) as previously published [30]. DOPC-GUVs were transferred into chambers with surface-immobilized insulin-loaded HDL particles and incubated for 5 min. For confocal measurements, GUVs enriched with His_6_-insulin linked to 18:1 DGS NTA (Ni) lipids were marked with an anti-insulin antibody conjugated to Alexa Fluor 488. The cross sections of GUVs were imaged, and the Membrane Enrichment Factor (MEF) was calculated using Equation (1), where µ_Membrane_ = mean intensity value of membrane area per cross section, and µ_Baseline_ = mean intensity of background:(1)$$\mathrm{MEF}=\frac{\mu_{{}_{\mathrm{Membrane}}}}{\mu_{{}_{\mathrm{Baseline}}}}$$

To test whether the study populations (125 μg/mL of HDL, 100 nM of insulin, and 125 μg/mL of insulin-loaded HDL) were different compared to the basal-level population (i.e., treatment with HBSS), the nonparametric Wilcoxon rank-sum test was performed [49].

## 3. Results

Insulin-loaded HDL particles were generated to test their proposed cargo delivery to cells. The insulin molecules were anchored to the outer membrane of the particles. The particles were characterized regarding size, topological shape, loading, and transfer and interaction properties in biomembranes. The HDL particles changed marginally in their topographical appearance (Figure 1). Their transfer and their stimulation properties in artificial (Figure 2) and living cell membranes (Figure 3) were studied.

### 3.1. Characterization of Insulin-Loaded HDL Particles

The topological characterization of insulin-loaded HDL particles with respect to HDL was performed using AFM measurements (Figure 1A,B). HDL particles were immobilized on a clean glass surface and subsequently imaged. The lateral size of each particle was analyzed. A statistical test for identifying different populations with respect to HDL particles was performed using the Kruskal–Wallis test. The results reveal that insulin-loaded HDL particles have a marginally increased size relative to native ones (Figure 1C,D). The height analyses of all HDL particle variants yielded the following values: (mean height ± error of mean) HDL = 13.5 nm ± 0.2 nm, insulin-loaded HDL = 16.2 nm ± 0.7 nm; # (HDL) = 130, and # (insulin-loaded HDL) = 14. Successful lipid-anchoring of insulin to HDL particles was tested using a fluorescently labeled antibody against insulin (see Appendix A).

### 3.2. Transfer of Lipid-Anchored Insulin from HDL Particles to Artificial Membranes

We addressed whether lipid-anchored insulin from HDL particles can be transferred to a target membrane. HDL particles were incubated with DGS-NTA lipids and subsequently purified. These modified HDL particles were immobilized via APTES to the support. Insulin-His_6_ was added, and then unbound molecules were removed by multiple washing steps. To demonstrate that HDL particles can transfer their lipid-anchored load to membranes solely via these particles, we added Giant Unilamellar Vesicles (GUVs) composed of DOPC lipids to the immobilized HDL particles. After the vesicles had settled, a transfer of lipid-anchored insulin previously linked to HDL particles occurred on the GUVs’ DOPC membrane (Figure 2B top). To visualize the transferred insulin molecules microscopically, a fluorescently labeled antibody against insulin was used (Figure 2A). An analysis (see method section) of the GUVs’ MEF was performed (Figure 2B bottom). The MEF of each analyzed GUV showed an enrichment of the GUVs’ membrane with insulin (MEF ± error of the mean =13.4 a.u. ± 1.2 a.u., Figure 2C).

Taken together, we observed physical transfer of insulin from HDL particles toward the GUV membranes after contact. Previous work [13,50] has demonstrated that LPs transfer/exchange their lipid load not only via receptor-mediated uptake but also via fusion with the target membrane. 

### 3.3. Cell Stimulation via Insulin-Loaded HDL Particles

We tested whether the transfer of lipid-anchored insulin exhibits a biological effect on CHO-K1-hIR/GLUT4-myc-GFP cells via monitoring the translocation of eGFP-GLUT4 to the cell membrane. In comparison, we triggered its translocation by adding soluble insulin or native HDL particles. For the estimation of the basal level, the cells were mock treated with Hanks’ balanced salt solution (HBSS). After stimulation of the cells with each sample, GPMVs were formed, and the mean intensity of translocated eGFP-GLUT4 was analyzed using confocal microscopy (Figure 3). This method ensured that only GLUT4 incorporated into the membrane was assessed, since those close to the cell membrane were no longer observable in GPMVs. To determine the stimulation efficiency, the mean fluorescence of the membranes was normalized to the basal level of HBSS. The following results were obtained: mean ± error of the mean (number of GPMVs analyzed): HBSS (basal level) = 100% (n = 478), 125 µg/mL of HDL = 98.0% ± 2.2% (n = 156), 100 nM of insulin = 127.9% ± 4.5% (n = 83), and 125 µg/mL of insulin-loaded HDL = 117.5% ± 2.0% (n = 328). The statistical analysis was performed using the Wilcoxon rank-sum test (alpha = 5%). With respect to the basal level, incubation with HDL particles showed no significant difference, whereas the positive control with soluble insulin and insulin-loaded HDL showed a significant increase in the mean fluorescent intensity (Figure 3C). To summarize, the results clearly show a significant increase in the translocation of eGFP-GLUT4 into the cell membrane after stimulation via insulin-loaded HDL particles, similar to the physiological response triggered by soluble insulin. Thus, various modes of translocation of GLUT4 are possible, either by lipid-transferred insulin or by insulin being anchored to HDL particles.

## 4. Discussion

Drug administration requires strategies, processes, and interaction knowledge regarding the absorption and transport of a drug in the body in order to achieve an ideal preventive or therapeutic effect. Approaches for drug delivery systems have advanced substantially, but conventional therapeutics are often hampered by insufficient drug concentrations at the pathological lesions, lack of cell-specific targeting, and various biobarriers [31,32,51]. Various natural nanoparticle systems have been utilized to serve as carriers for specific drugs, DNA expression vectors, RNAs, or nutritional supplements. Their unique properties regarding size and distribution, functional surface groups, high payload capacity, and ability to trigger drug release allow researchers to tailor them to the specific requirements of a drug. To avoid off-target effects and improve efficiency, a high cell specificity is essential. In addition, low toxicity of the vehicles themselves is among the most important requirements in the development of nanoscopic carriers.

In this work, HDL particles were utilized as drug carriers because of their main advantages in terms of biocompatibility/degradability and characteristics (i.e., size) as defined by apolipoprotein(s). Hence, hydrophilic biomolecules were anchored to their shell. The novelty of our work includes (i) our method of incorporation of a drug (i.e., his-tagged insulin via DGS-NTA lipid) to an otherwise unaltered, natural nanobioparticle (i.e., HDL particle), and (ii) our methods of particle characterization (i.e., AFM and FM) and biological effect measurement (i.e., confocal imaging of GPMVs). This study focuses on the characterization of the assembled particles, and their uptake and interaction with biomembranes. As an example for a transported drug, insulin was chosen. Insulin was bound to the HDL membrane via lipid anchorage and interaction with artificial membrane, and the response at a cellular level were studied. The conclusion that successful anchoring of insulin to HDL particles via lipids could be performed is supported by the marginal but statistically significant increase in their shape (Figure 1). Surfaces with different densities of immobilized insulin-loaded HDL particles were offered (Appendix A). In this regard, due to the doubled density of the loaded HDL particles, we also expected a doubling of the transferred insulin molecules. Both surfaces yielded saturation of transferred lipid-anchored insulin molecules from the HDL particles to the contacting GUV membrane. Consequently, the same amount of DGS NTA(Ni)-His_6_-insulin was transferred to the target membrane within 10 min, regardless of the insulin density offered. In past studies [52,53], as well as in the membrane experiments shown (see Figure 2 and Figure 3), HDL particles exchange their lipids with the target membrane. It can be concluded from these observations that insulin is also transferred to the target membrane with its lipid anchoring. Thus, a question arises regarding how lipid-anchored insulin stimulates GLUT4 transport. HDL particle-anchored insulin may interact directly with insulin receptor (i.e., particle/cell membrane interaction). Additionally, lipid-anchored insulin in the cell membrane may have the steric ability to interact with its receptor or bind to the receptor of a neighboring cell. To determine whether insulin anchored to HDL particles causes stimulation of GLUT4, insulin was directly immobilized on a spherical AFM tip and guided to the cell membrane. In this way, we generated local stimulation at the target membrane with an insulin-loaded nanopipette. An increase in the translocation of GLUT4-eGFP was observed after the local stimulation via a biotin-insulin tip (Appendix A). Thus, we conclude that local stimulation is possible, and insulin interacts directly with the cell membrane. Therefore, two stimulation effects within a cell matrix may occur, either via direct insulin-loaded HDL particles or via transferred lipid-anchored insulin from the particle to the target membrane.

In this study, HDL particles were altered to understand and improve the delivery of therapeutic agents to biomembranes This work demonstrates the successful loading of HDL particles with insulin, as well as the initiated cellular stimulation of GLUT4 transporters. Their topographical structure changes only slightly due to the incorporation of lipid-anchored molecules. Likewise, the physiological transport pathway is not expected to be affected.

## Figures and Tables

**Figure 1 membranes-13-00471-f001:**
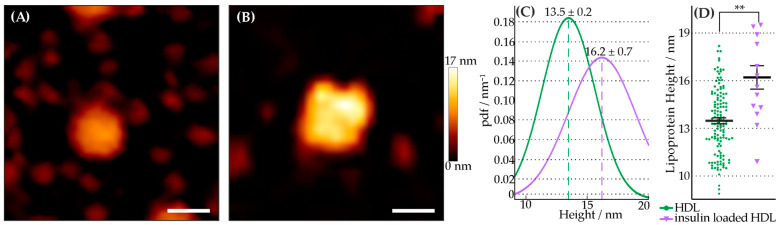
Topographical analysis of HDL particle variants. (**A**,**B**) show representative topographical images of a HDL (**A**) and an insulin-loaded HDL particle (**B**) (same color scale, scale bar = 20 nm). The images were obtained using JPK BioAFM-NaonWizard 4 with the quantitative imaging mode (QI™ mode) at a maximal force of 500 pN using a FASTSCAN-D cantilever with a nominal spring constant of 0.25 N/m and a nominal tip apex radius of 5 nm. (**C**) shows the probability density function (pdf) of the measured height for HDL (green) and insulin-loaded HDL (violet) particles. (**D**) Scatter plot and error bar (mean ± error of mean) of HDL (green) particles’ and insulin-loaded HDL (violet) particles’ height. Each data point (circle, triangle) represents the height of an individual particle. The results of the Kruskal–Wallis test are indicated by connecting lines. (** *p* < 0.01); number of analyzed HDL particles per experiment: # (HDL) = 130, and # (insulin loaded HDL) = 13.

**Figure 2 membranes-13-00471-f002:**
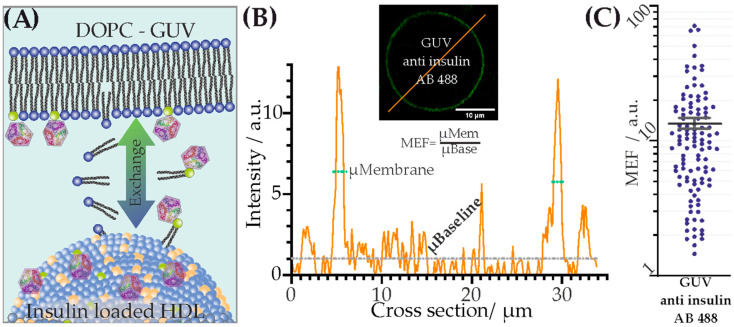
Interaction of insulin-loaded HDL particles with artificial membranes. (**A**) Schematic of the lipid exchange process of His_6_-insulin linked to 18:1 DGS NTA(Ni) lipids. Immobilized insulin-loaded HDL particles exchange lipids with the target membrane (DOPC–GUV). (**B**) Top inset depicts a representative confocal image of a DOPC–GUV enriched with fluorescent anti-insulin antibodies bound to His_6_-insulin linked to 18:1 DGS NTA(Ni) lipids. The orange line through the GUV depicts the location of intensity cross-sectional measurement. The bottom graph shows fluorescence intensity (orange line in top inset) of anti-insulin antibody labeled with Atto488 binding to His_6_-insulin linked to 18:1 DGS NTA(Ni) lipids. (**C**) shows the scatter plot and error bar (mean ± error of mean) of MEF for every analyzed GUV membrane (N = 111). The calculation of Membrane Enrichment Factor (MEF) was performed via an analysis of the cross sections of GUV membranes (see exemplary image shown in (**B**)). MEF represents the ratio between mean membrane (green dotted line, µ_Mem_) and mean baseline (gray dotted line, µ_Base_) intensities. Mean value ± error of the mean =13.4 a.u. ± 1.2 a.u. (each individual dot in (**C**) represents a single GUV-MEF).

**Figure 3 membranes-13-00471-f003:**
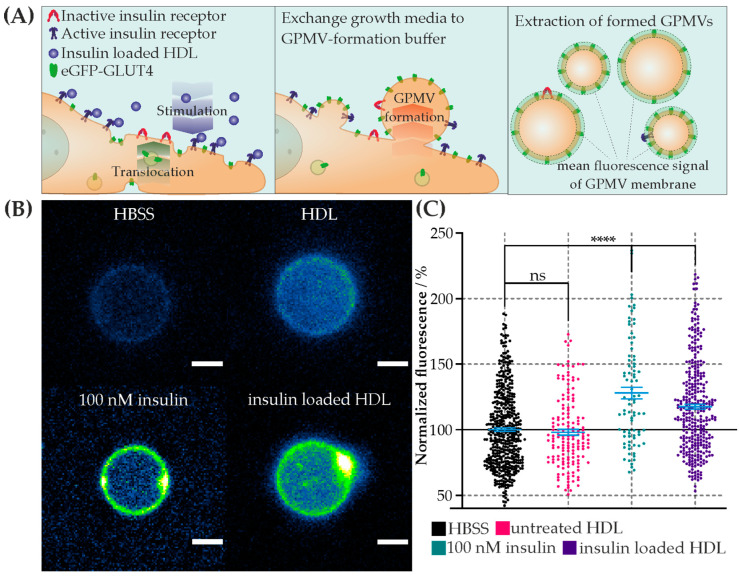
Interaction of insulin-loaded HDL particles with eGFP-GLUT4-expressing cells. (**A**) Schematic representation of the experimental approach to determine stimulation effect. (**Left**) Interaction and stimulation of insulin receptors of living eGFP-GLUT4-expressing cells with insulin-loaded HDL particles. The stimulation results show the translocation of intracellular eGFP-GLUT4 into the cell membrane. (Middle) Incubation with Giant Plasma Membrane Vesicle (GPMV)-formation buffer and resulting formation of eGFP-GLUT4-translocated GPMVs on the cell membrane. (**Right**) Extracted eGFP-GLUT4-containing GPMVs and analysis of the fluorescence signal. (**B**) Representative confocal microscopic images of GPMVs formed from cells incubated with HBSS (**left top**), HDL particles (**right top**), 100 nM of insulin (**left bottom**), and insulin-loaded HDL particles (**right bottom**) show a base and increased signal of eGFP-GLUT4 in the membrane (equal color scaling, scalebar 2 µm). (**C**) Scatter plot and error bar (mean ± error of mean) of normalized mean eGFP-GLUT4 signal per GPMV membrane for each stimulation condition in %. Hanks’ balanced salt solution (HBSS) data were used for normalization. Stimulation conditions: HBSS—basal level (black); 125 µg/mL of HDL particles (magenta); 100 nM of insulin (green); 125 µg/mL of insulin-loaded HDL particles (violet). Mean value ± error of the mean: I_HBSS_ = 100%, I_HDL_ = 98.0% ± 2.2%, I_100 nM insulin_ = 127.9% ± 4.5%, and I_insulin loaded HDL_= 117.5% ± 2.0%. The test results of Wilcoxon rank-sum test compared to the HBSS results are indicated by connecting lines. (**** *p* < 0.0001, n.s. = no significant difference *p* > 0.05). Number (N) of analyzed GPMVs per stimulation condition: N_HBSS_ = 478, N_100 nM insulin_ = 83, N_HDL_ = 156, and N_insulin loaded HDL_ = 328.

## Data Availability

The data presented in this study are available from the corresponding author upon request. The data are not publicly available due to restrictions of the involved scientists and facilities.

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
