# Peer review of "Lipoprotein Particles as Shuttles for Hydrophilic Cargo"

_membranes, 2023, doi:10.3390/membranes13050471_

Round 1

Reviewer 1 Report

This paper describes the insulin-loaded GUV from the insulin-loaded HDL by the lipid exchange.  While this paper contributes to the medical field, I have some reservations that should be addressed before publication.  My comments are as follows.

1. What percent of the insulin-loaded HDL was incorporated into the GUV?  And what percent of the insulin-loaded HDL to the total number of DOPC composing GUV?

2. Can the insulin loaded on the DOPC have its inherent function?  Or can the insulin easily separate from the DOPC to function?

3. What advantage of the insulin-loaded HDL over using micelles and vesicles that load insulin without binding?

The authors should clarify the above problems to promote a better understanding of readers.

Reviewer 2 Report

The submitted article entitled “Lipoprotein particles as shuttle for hydrophilic cargo” by Weber et al., presents a highly focused manuscript on utilizing human isolated HDL particles for uptake and delivery of insulin in target membranesInterestingly, a cellular assay was used to demonstrate biological effects through the translocation of eGFP-GLUT4, which is internal to the cell membrane. Previous studies have used similar protocols and biophysical techniques to look at different forms of HDL nanoparticles impact on model systems to include in vitro and in vivo studies.  These previous studies include the use of DGS-NTA lipids for immobilizing and delivery of biologicals. Overall, the article is very well written, and the authors have provided detailed information and statistical analysis of the data. All the figures reflect the authors findings and conclusions. I believe this is an important area of research and the general readers will be interested in the manuscript and its findings related to novel techniques for natural nanoparticle systems. I would recommend this paper for publication after addressing several specific comments (see below).

Specific Comments:

1.     Both natural and reconstituted HDLs have been used for hydrophobic molecule loading and delivery (Fox CA, Moschetti A, Ryan RO. Biochim Biophys Acta Mol Cell Biol Lipids. 2021). Some mention of the difference between native and reconstituted HDLs is worth mentioning in the introduction. The authors may also need to further demonstrate that their isolation process still contains all the protein and lipid constituents that may define differences between spherical and disc shaped native and reconstituted HDLs. One could argue that the addition of the DGS-NTA lipids means the authors are really working with a rHDL particle and full characterization might be warranted. 

2.     The following papers should be considered for citation given they demonstrated the use of DGS-NTA lipids for delivery systems for uptake of biologicals. The second manuscript focuses specifically on utilizing DGS-NTA lipids within rHDL particles. 

a.     Herringson and Altin, Convenient targeting of stealth siRNA-lipoplexes to cells with chelator lipid-anchored molecules. 2009. DOI: 10.1016/ PMID: 19595724

b.     Fischer et al., Immobilization of His-tagged proteins on nickel-chelating nanolipoprotein particles. 2009. Bioconjugate Chem. DOI: 10.1021/bc8003155. PMID: 19239247

3.     The use of biophysical techniques such as AFM and confocal microscopy for characterizing multiple types of nanoparticles to include different forms of HDLs and GUVs are well documented withing the published literature. The manuscript would benefit by including additional citations for previously published work beyond the authors own group.

Round 2

Reviewer 2 Report

The submitted article entitled “Lipoprotein particles as shuttle for hydrophilic cargo” by Weber et al., presents a highly focused manuscript on utilizing human isolated HDL particles for uptake and delivery of insulin in target membranes. Interestingly, a cellular assay was used to demonstrate biological effects through the translocation of eGFP-GLUT4, which is internal to the cell membrane. Previous studies have used similar protocols and biophysical techniques to look at different forms of HDL nanoparticles impact on model systems to include in vitro and in vivo studies.  These previous studies include the use of DGS-NTA lipids for immobilizing and delivery of biologicals. Overall, the article is very well written, and the authors have provided detailed information and statistical analysis of the data. All the figures reflect the authors findings and conclusions. I believe this is an important area of research and the general readers will be interested in the manuscript. I would recommend this paper for publication in its current form.